# Free and Immobilized Lecitase™ Ultra as the Biocatalyst in the Kinetic Resolution of (*E*)-4-Arylbut-3-en-2-yl Esters

**DOI:** 10.3390/molecules25051067

**Published:** 2020-02-27

**Authors:** Aleksandra Leśniarek, Anna Chojnacka, Radosław Drozd, Magdalena Szymańska, Witold Gładkowski

**Affiliations:** 1Department of Chemistry, Wroclaw University of Environmental and Life Sciences, Norwida 25, 50-375 Wroclaw, Poland; anna.chojnacka@upwr.edu.pl; 2Department of Microbiology and Biotechnology, Faculty of Biotechnology and Animal Husbandry, West Pomeranian University of Technology, Szczecin, 45 Piastów Avenue, 71-311 Szczecin, Poland; rdrozd@zut.edu.pl (R.D.); magdalena.szymanska@zut.edu.pl (M.S.)

**Keywords:** (*E*)-4-arylbut-3-en-2-ols, enantioselective hydrolysis, kinetic resolution, Lecitase™ Ultra, immobilization, cyanogen bromide-activated agarose

## Abstract

The influence of buffer type, co-solvent type, and acyl chain length was investigated for the enantioselective hydrolysis of racemic 4-arylbut-3-en-2-yl esters using Lecitase™ Ultra (LU). Immobilized preparations of the Lecitase™ Ultra enzyme had significantly higher activity and enantioselectivity than the free enzyme, particularly for 4-phenylbut-3-en-2-yl butyrate as the substrate. Moreover, the kinetic resolution with the immobilized enzyme was achieved in a much shorter time (24–48 h). Lecitase™ Ultra, immobilized on cyanogen bromide-activated agarose, was particularly effective, producing, after 24 h of reaction time in phosphate buffer (pH 7.2) with acetone as co-solvent, both (*R*)-alcohols and unreacted (*S*)-esters with good to excellent enantiomeric excesses (ee 90–99%). These conditions and enzyme were also suitable for the kinetic separation of racemic (*E*)-4-phenylbut-3-en-2-yl butyrate analogs containing methyl substituents on the benzene ring (**4b**,**4c**), but they did not show any enantioselectivity toward (*E*)-4-(4’-methoxyphenyl)but-3-en-2-yl butyrate (**4d**).

## 1. Introduction

Lecitase™ Ultra (LU) is a commercially available chimeric enzyme that is the product of the fusion of lipase genes from *Thermomyces lanuginosus* and phospholipase A_1_ genes from *Fusarium oxysporum* [1]. This enzyme combines the stability of lipase and activity of phospholipase A_1_ and was initially designed for the degumming of plant oils during rafination [2,3,4,5,6]. In this process, phospholipids contained in the oil phase are hydrolyzed by Lecitase™ Ultra at the *sn*-1 position to lysophospholipids, which migrate to the aqueous phase. Since then, this fundamental reaction catalyzed by Lecitase™ Ultra has also found other food and none-food applications i.a. hydrolysis of soy phosphatidylcholine to produce lysophosphatidylcholine [7,8] and α-glycerophosphocholine [9,10], modification of flour in bread making [11,12,13], and rafination of oils used in the production of biodiesel [14,15]. Except for hydrolytic activity, Lecitase™ Ultra is also able to catalyze esterification, acidolysis, and alcoholysis, which has found application in the production of oils enriched with omega-3 fatty acids [16] as well as in the production of diglycerides [17,18,19,20] and structured phospholipids [21,22,23,24,25,26,27]. Due to its wide enzymatic activity, the application of Lecitase™ Ultra has been further extended to the production of flavor low-chain esters i.a. methyl benzoate and methyl butanoate [28] and regioselective hydrolysis of peracetylated mono- and disaccharides to produce key intermediates in the preparation of different glycoderivatives such as oligosaccharides, glycolipids, or glycopeptides [29].

One of the most popular uses of lipases in organic chemistry is the kinetic resolution of racemic mixtures [30,31,32]. Lecitase™ Ultra has been also employed for this goal and there are few reports concerning the application of Lecitase™ Ultra-catalyzed enantioselective hydrolysis of esters for the production of some important chiral drug intermediates. These applications covered the resolution of esters of 2-hydroxyacids with a phenyl ring including mandelates [33,34,35,36,37,38], resolution of glycidate esters [35,39,40], and *N*-acetyl-α-amino acid methyl esters [35]. Another example of an application in asymmetric synthesis was Lecitase™ Ultra-catalyzed regioselective hydrolysis of prochiral dimethyl 3-phenylglutarate [41].

Immobilization of enzymes on different supports is a useful tool to increase their stability, activity, and selectivity under operational conditions [42,43,44]. Due to a simpler separation of the biocatalyst from the reaction environment, such preparations may be reused in many catalytic cycles without losing their catalytic properties, which increases the economic efficiency of the industrial process [45]. However, in many cases, it is also related with a decrease in enzyme activity caused by various types of distortion, diffusion, and steric problems [46]. As Lecitase™ Ultra is commercially available only in its free form as the water solution, many attempts have been made to prepare an immobilized form of this enzyme for different purposes, for example, by entrapment in gelatin hydrogel [47] or in different calcium alginates [48], immobilization on the derivatives of cellulose [49,50], polystyrene resin [51], styrene-divinylbenzene beads [37,52], magnetic nanoparticles [53], or by encapsulation in AOT(bis(2-ethylhexyl) sulfosuccinate sodium salt)/isooctane reverse micelles [54]. In the processes of the kinetic resolution of racemates catalyzed by immobilized Lecitase™ Ultra, the best results were obtained using gelatin [35,39], various derivatives of agarose [33,36,38,41,55], hexyl- and butyl-toyopearl [34], or epoxy activated polymer Dilbead VWR coated with polyethylenimine (PEI) and crosslinked with glutaraldehyde [40].

In previous investigations, kinetic resolutions of racemic esters through their Lecitase™ Ultra-catalyzed enantioselective hydrolysis has been limited mainly to the esters of chiral carboxylic acids [34,35,39,41]. However, since a large variety of chiral esters possess the stereogenic center in the nucleophilic part of molecule, to extend the substrate scope and increase the usability of this enzyme in the asymmetric synthesis, we employed Lecitase™ Ultra for the biocatalytic resolution of racemic 4-arylbut-3-en-2-ol esters. In our working group, these alcohols are the subject of special interest as chiral precursors of the optically active lactones with antiproliferative activity [56,57]. Likewise, they have been used in the production of chiral drugs [58,59].

In our previous paper, we reported the preliminary results of the free Lecitase™ Ultra-mediated kinetic resolution of model racemic esters of (*E*)-4-phenylbut-3-en-2-ol through their enantioselective hydrolysis, achieving moderate to good enantiopurity of the products [60]. Here, we present the results of our further investigations in this field, aiming to increase the process efficacy through optimization of some reaction conditions (pH of the reaction medium, addition of organic co-solvent) as well as by enzyme immobilization. To determine the substrate specificity and extend the application of Lecitase™ Ultra, the hydrolysis of some analogs of (*E*)-4-phenylbut-3-en-3-ol esters differing in aromatic fragment of the molecule was also studied.

## 2. Results and Discussion

Eleven known (**2a–d, 3a–d** and **4a,b,d**) and one new (**4c**) racemic allyl esters were obtained in high yields (96–98%) by esterification of corresponding known [61,62,63] (*E*)-4-arylbut-3-en-2-ols (**1a–d**) with appropriate acid chlorides, according to the standard procedure (Scheme 1).

Studies toward the kinetic resolution of the prepared compounds were started for the free Lecitase™ Ultra (LU)-catalyzed hydrolysis of model 4-phenylbut-3-en-2-ol esters (**2a–4a**) to investigate the influence of the length of an acyl chain, type of co-solvent, and type of buffer on the process stereoselectivity (Table 1).

In previous studies, the enantioselective hydrolysis of carboxylic acid esters catalyzed by LU was carried out in a phosphate buffer [33] or Tris-HCl buffer [35], therefore in our first experiments, we employed these two reaction media. (*E*)-4-Phenylbut-3-en-2-ol esters with different acyl chain lengths (**2a–4a**) were subjected to hydrolysis at room temperature, with acetone as a co-solvent. In all cases, a higher enantioselectivity of hydrolysis was noticed in the reactions carried out in the phosphate buffer (Entries 4–6). Higher enantiomeric excess of alcohol **1a** and a shorter reaction time necessary to achieve the best resolution was observed with the elongation of the acyl chain. The best results were obtained for butyrate **4a**; in this case, the highest enantioselectivity (E > 200) was observed after 96 h of reaction, resulting in the optically pure product (Entry 6). The positive effect of the acyl chain length of hydrolyzed esters **2a–4a** on their kinetic resolution was a constant tendency, observed also in experiments carried out in Tris-HCl (Entries 1–3) as well as using other co-solvents (Entries 7–9 and 10–12). These investigations confirm the high specificity of Lecitase™ Ultra toward esters of butyric acid, which was earlier found by Mishra et al. [35]. In their experiments, Lecitase™ Ultra did not accept glycerol triacetate as the substrate but exhibited hydrolytic activity toward glycerol tributyrate.

The influence of the co-solvent on the kinetic resolution of esters **2a–4a** was determined for the reactions carried out in the phosphate buffer. For each of the substrates, the results of the reactions with acetone as the co-solvent were compared with those carried out with the addition of dimethylformamide (DMF) and dimethyl sulfoxide (DMSO). In all cases, the highest enantioselectivity was observed when the co-solvent was acetone (Entries 4–6). In the hydrolysis of propionate **3a** with the addition of acetone (Entry 5) and DMF (Entry 8), the enantiomeric excesses of both unreacted ester and the produced alcohol were comparable, while the addition of DMSO was the least effective (Entry 11). In the case of the hydrolysis of acetate **2a**, the highest enantiomeric purity of both products was observed for the reaction with the addition of acetone (Entry 4), while the addition of DMSO resulted in slightly better results than the addition of DMF (Entries 7 and 10). Significant impact of the co-solvent on the enantioselectivity of the reaction was observed for the hydrolysis of butyrate **4a**. In this case, a pure enantiomer of alcohol **1a** was produced only in the presence of acetone (Entry 6); the addition of DMF (Entry 9) and DMSO (Entry 12) resulted in a decrease of *ee* for alcohol **1a** to 87–88%. On the other hand, the highest *ee* of unreacted butyrate **4a** (69%) was noticed using DMSO as the co-solvent (Entry 12).

In subsequent experiments, free Lecitase™ Ultra was applied to the hydrolysis of esters of (*E*)-4-phenylbut-3-en-2-ol analogues differing in the structure of the aromatic fragment (Table 2).

Based on the previous results of the kinetic resolution of esters **2a–4a**, the procedure using phosphate buffer as the reaction medium and the addition of acetone as a co-solvent was applied. Regardless of the acyl chain length, esters of (*E*)-4-(4′-methoxyphenyl)but-3-en-2-ol (**2d–4d**) were hydrolyzed with no enantioselectivity (Entries 7–9), and for esters containing methyl groups (**2b–4b, 2c–4c**) lower enantioselectivities of the reactions were observed (E ≤ 18) compared to (*E*)-4-phenylbut-3-en-2-yl esters **2a–4a** (Table 1, entries 4–6). The beneficial effect of the elongation of the acyl chain on the results of kinetic resolution was confirmed only in the group of esters with the *p*-methyl substituted phenyl ring (**2b–4b**). Better resolution was observed for compounds with a 2,5-dimethylsubstituted phenyl ring than for those with the *p*-methylphenyl substituent, which was particularly noticeable in the case of acetates and propionates.

The positive effect of immobilization on the activity and enantioselectivity of enzymatic reactions is known from the literature [42]. In the case of LU, this relationship was confirmed in the hydrolysis of α-hydroxy acids esters. The highest increase of enantioselectivity, from E = 4 for the free enzyme to E = 19.5 for the enzyme immobilized in gelatin was observed for the hydrolysis of ethyl 2-hydroxy-4-oxo-4-phenylbutyrate, resulting in optically pure unreacted ester at 60–65% conversion [35]. Taking this into consideration, in our further studies, immobilized preparations of Lecitase™ Ultra were used as biocatalysts of the kinetic resolution of esters of alcohols with a 4-arylbut-3-en-2-ol system. For this purpose, the commercial enzyme was immobilized on four carriers: calcium alginate; Supelite™ DAX-8 acrylic resin; cyanogen bromide-activated agarose; and modified bacterial cellulose. The activity of immobilized preparations (Table 3) was measured using *p*-nitrophenyl palmitate as a substrate [64]. The highest, comparable activity was determined for the preparations obtained after immobilization on cyanogen bromide-activated agarose (LU-CNBr) and acrylic resin (LU-DAX). Preparation LU-CNBr was previously used in the production of optically pure (*S*)-3-methyl phenylglutarate through the asymmetric hydrolysis of dimethyl 3-phenylglutarate (48 h, 80% yield) [41].

(*E*)-4-Phenylbut-3-en-2-yl acetate (**2a**), (*E*)-4-phenylbut-3-en-2-yl propionate (**3a**), and (*E*)-4-phenylbut-3-en-2-yl butyrate (**4a**) were subjected to kinetic resolution with immobilized preparations of LU and the results were compared with those obtained previously for the free enzyme (Figure 1, Figure 2 and Figure 3).

One can see that the use of LU-CNBr and LU-MBC significantly shortened the time needed to achieve the kinetic resolution of (*E*)-4-phenylbut-3-en-2-yl acetate (**2a**) up to 24 h and 48 h, respectively (Figure 1).

Likewise, the use of these preparations also had a positive effect on the parameters of kinetic resolution. The most spectacular improvement in the enantioselectivity of reaction (E > 200) was achieved using the LU-MBC preparation (Figure 1). In this case, very high enantiomeric excesses of alcohol **1a** and unreacted acetate **2a** were achieved of 97% and 96%, respectively. A highly enantioselective (E = 99) was also the reaction catalyzed by the LU-CNBr, in this case, after 24 h, the conversion exceeded 50%, which resulted in a lower *ee* of alcohol **1a** (90%), but excellent *ee* of acetate **2a** (99%). At a shorter reaction time (20 h, data not shown), when the conversion was close to 50%, the *ee* of alcohol **1a** was higher, but a lower *ee* of acetate **2a** was observed. The process of immobilization of the enzyme on Supelite™ DAX-8 acrylic resin and in calcium alginate did not bring the expected results. For the reaction catalyzed by LU-DAX, a slight increase of enantioselectivity was observed, affording alcohol **1a** with 76% *ee*. However, the only 20% conversion degree achieved in this reaction resulted in significantly lower enantiomeric excess of unreacted ester **2a** compared to the reaction catalyzed by free LU. Very low enantioselectivity of hydrolysis was observed in the case using Lecitase™ Ultra Alginate (LU-ALG).

Preparations LU-CNBr and LU-MBC were also the most efficient biocatalysts for the hydrolysis of propionate **3a** (Figure 2).

The highest increase in the enantioselectivity of the reaction (E = 170) compared to the reaction catalyzed by the free enzyme was observed for LU-CNBr. Using this preparation allowed us to obtain enantiomerically pure propionate **3a** and enantiomerically enriched alcohol **1a** (*ee* 94%) after 24 h. The LU-MBC preparation hydrolyzed propionate **3a** with an enantioselectivity comparable to that observed for the free LU-catalyzed reaction, but at a much higher rate. Accordingly, the enantiomers of the substrate were resolved in a significantly shorter time (24 h) and a 53% conversion of alcohol **1a** and unreacted ester **3a** was obtained with enantiomeric excesses of 79% and 90%, respectively. In the processes catalyzed by LU-DAX and LU-ALG, both the enantioselectivity of the reaction and the optical purities of unreacted propionate **3a** and alcohol **1a** were lower than those observed in free LU-catalyzed hydrolysis.

Particularly noteworthy are the results of the hydrolysis of (*E*)-4-phenylbut-3-en-2-yl butyrate (**4a**) (Figure 3).

For each preparation, the enzyme immobilization significantly increased its hydrolytic activity toward the substrate. Using preparations LU-CNBr, LU-DAX, and LU-ALG, kinetic resolution was achieved just after 24 h, affording alcohol **1a** with *ee* in the range of 94–97% and unreacted ester **4a** with 95–96% *ee*. A lower enantiomeric excess of alcohol **1a** (*ee* 83%) was observed only when LU-MBC preparation was used; in this case, after 24 h, the conversion exceeded 50% and it was possible to obtain optically pure unreacted butyrate **4a**.

Enantiomeric purities of the products of the kinetic resolution of racemic (*E*)-4-phenylbut-3-en-2-yl esters through Lecitase™ Ultra-catalyzed hydrolysis were comparable or higher than those obtained previously involving commercially available lipase preparations. Among the different lipases tested by Ghanem et al. [65], the most effective biocatalyst of the kinetic resolution of acetate **2a** was lipase B from *Candida antarctica* (CALB). Optically pure alcohol **1a** and unreacted acetate **2a** with only 80% *ee* were obtained after 24 h of reaction in the phosphate buffer (pH 6.0) with the addition of toluene as the co-solvent. Similarly, in our earlier work, CALB was used to resolve racemic acetate **2a** and propionate **3a** in the reaction conditions analogous to those used in this study. A short time was needed to achieve kinetic resolution (6 h and 2 h, respectively), but significantly lower enantiomeric excess of unreacted ester was observed (80–85%) [60]. In other work, Thalén et al. [66] used CALB in the hydrolysis of enantiomerically enriched (*S,E*)-4-phenylbut-3-en-2-yl butyrate (*ee* 95%) to obtain optically pure butyrate **4a** after 2 h of reaction carried out in a phosphate buffer (pH 7.2), but the reaction was carried out at high temperature (60 °C).

Taking into consideration that the most effective biocatalyst in the kinetic resolution of all 4-phenylbut-3-en-2-yl esters **2a–4a** was LU-CNBr, it was decided to extend its scope of application to the kinetic resolution of butyrates **4b**, **4c**, and **4d** containing the substituted benzene ring (Table 4), which have been hydrolyzed previously with free enzyme (Table 2).

The preparation was an effective biocatalyst for the enantioselective hydrolysis of butyrate with the *p*-methylphenyl (**4b**) and 2,5-dimethylphenyl (**4c**) substituent. Significantly higher reaction rates and enantioselectivities were observed compared to the reactions catalyzed by free LU. Enantiomerically enriched alcohols (98% *ee* for **1b** and 99% *ee* for **1c**) and unreacted esters (89% *ee* for **4b** and 99% *ee* for **4c**) were obtained after 24 h or 48 h. On the other hand, the use of LU-CNBr did not affect the enantioselectivity of the hydrolysis of butyrate with the *p*-methoxyphenyl substituent (**4d**) and, similar to the reaction catalyzed by the free enzyme, both the alcohol **1d** and unreacted ester **4d** were racemic mixtures.

The aim of our last experiment was to determine the minimal enzyme dosage required for the effective kinetic resolution of the tested substrate in a 24 h reaction (Figure 4). The substrate of this reaction was (*E*)-4-phenylbut-3-en-2-yl acetate (**2a**) and hydrolysis was carried out using 100 mg of the starting ester. Regardless of an enzyme dosage, the enantioselectivity of reaction was consistently high (E > 200), but significant decrease of the conversion and the optical purity of unreacted ester **2a** was observed when the enzyme dosage was lower than 0.03 U.

In order to prove the enantiopreference of Lecitase™ Ultra toward 4-arylbut-3-en-2-yl esters, alcohols and unreacted esters obtained after hydrolysis of butyrates **4a–4c** by LU-CNBr were separated using preparative TLC and the configurations of known isolated isomers were unambiguously confirmed by comparing the signs of their specific rotation with the data in the literature (see Materials and Methods, paragraph 3.9.3). Similar to most lipases, Lecitase™ Ultra shows enantiopreference according to Kazlauskas’ rule, hydrolyzing (*R*)-enantiomers of tested substrates with a higher reaction rate, which results in the production of (*R*)-alcohols and (*S*)-enantiomers of unreacted esters.

## 3. Materials and Methods

### 3.1. Enzyme and Chemicals

Lecitase™ Ultra (LU, >10 000 U × g^−1^) was obtained from Sigma-Aldrich (St. Louis, MO, USA). Racemic alcohols: (E)-4-phenylbut-3-en-2-ol (1a), (E)-4-(4′-methylphenyl)but-3-en-2-ol (1b), (E)-4-(2′,5′-dimethylphenyl)but-3-en-2-ol (1c), and (E)-4-(4′-methoxyphenyl)but-3-en-2-ol (1d) were synthesized according to the procedures described earlier [61,62,63]. Acetyl chloride (≥99%), propionyl chloride (98%), butyryl chloride (≥99%), sodium alginate, calcium chloride (≥96%), p-nitrophenyl palmitate (*p*-NPP), p-nitrophenol (*p*-NP), and polyethyleneimine (PEI) were purchased from Sigma-Aldrich (St. Louis, MO, USA). Cyanogen bromide-activated agarose (Sepharose^®^ 4B) was purchased from GE Healthcare Bio-Sciences AB (Uppsala, Sweden). Supelite™ DAX-8 was purchased from Supelco Analytical (Bellefonte, PA, USA). Sodium alginate was purchased from Sigma-Aldrich. Other chemicals were of analytical grade.

### 3.2. Analysis

Racemic esters **2a–d**, **3a–d**, and **4a–d** were analyzed on an Agilent Technologies 6890N gas chromatograph (Santa Clara, CA, USA) equipped with hydrogen as the gas carrier, autosampler, split injection (50:1), and a flame ionization detector (FID) detector. Enantiomeric purity of the compounds was determined using a CP(Chrompack)-Chiral-Dex CB column (25 m × 0.25 mm × 0.25 m, Varian, Palo Alto, CA, USA). The temperature program for the products of hydrolysis of acetates 2a–d and propionates 3a–d are as follows: injector 280 °C, detector (FID) 280 °C, column 80–130 °C (0.5 °C × min^−1^), 130–200 °C (30 °C min^−1^), 200 °C (2 min); for the products of hydrolysis of butyrate 4a: injector 280 °C, detector (FID) 250 °C, column 80–120 °C (1 °C min^−1^), 120–140 °C (0.5 °C min^−1^); for the products of hydrolysis of butyrate 4b: injector 230 °C, detector (FID) 250 °C, column 40–160 °C (0.5 °C min^−1^), 160–190 °C (5 °C min^−1^); for the products of hydrolysis of butyrate 4c and 4d: injector 280 °C, detector (FID) 250 °C, column 40–140 °C (0.3 °C min^−1^), 140–180 °C (1 °C min^−1^). Analytical thin layer chromatography (TLC) was carried out on silica gel coated aluminum plates (DC-Alufolien Kieselgel 60 F_254_, Merck, Darmstadt, Germany). Compounds were visualized by spraying the plates with solution of 1% Ce(SO_4_)_2_ and 2% H_3_[P(Mo _3_O_10_)_4_] in 10% H_2_SO_4_. Silica gel (Kieselgel 60, 230–400 mesh, Merck) was used as a stationary phase in the chromatographic purification of synthesized esters **2a–d**, **3a–d**, and **4a–d**. Products of kinetic resolution were separated on preparative TLC silica gel glass plates (Uniplate™ UV_254_, layer thickness 1000 μm, 20 cm × 20 cm, Analtech, Newar, DE, USA). The optical rotations were measured on a Jasco P-2000-Na digital polarimeter (Easton, PA, USA) with an intelligent Remote Module (iRM) controller. The NMR spectra (^1^H NMR, ^13^C NMR, HMBC, and HMQC) were recorded on a Bruker Avance II 600 MHz spectrometer (Bruker, Rheinstetten, Germany). Samples were dissolved in CDCl_3_ and chemical shifts were referenced to residual solvent signals (δH = 7.26, δC = 77.0). The IR spectra were recorded on a Mattson IR 300 Thermo Nicolet spectrophotometer (Mattson, Waltham, MA, USA). High resolution mass spectrum (HRMS) of 4-(2′,5′-dimethylphenyl)but-3-en-2-yl butyrate (4c) was recorded on a Waters ESI-QTOF Premier XE spectrometer (Waters Corp., Millford, MA, USA) using the electron spray ionization (ESI) technique. The index of refraction was measured on an Abbe refractometer (Carl Zeiss, Jena, Germany).

### 3.3. Synthesis of Racemic Esters **2a**–**d**, **3a**–**d**, and **4a**–**d**—General Procedure

Alcohol **1a**–**d** (14 mmol) was dissolved in 100 mL of dry diethyl ether and 10 mL of pyridine. The mixture was stirred in an ice bath and 19 mmol of corresponding acylating agent (acetyl chloride, propionyl chloride, or butyryl chloride) was added dropwise. The reaction was continued at room temperature until the alcohol reacted completely (24 h, TLC). The reaction mixture was acidified with 1M HCl, the product was extracted with diethyl ether (3 × 40 mL) and separated by silica gel column chromatography (hexane:acetone, 10:1) to afford pure ester.

(*E*)-4-phenylbut-3-en-2-yl acetate (**2a**): Yield 96% (2.7 g), spectroscopic data consistent with those reported in the literature [67].

(*E*)-4-(4′-methylphenyl)but-3-en-2-yl acetate (**2b**): Yield 98% (2.6 g), spectroscopic data consistent with those reported in the literature [68].

(*E*)-4-(2′,5′-dimethylphenyl)but-3-en-2-yl acetate (**2c**): Yield 97% (2.8 g), spectroscopic data consistent with those reported in the literature [63].

(*E*)-4-(4’-methoxyphenyl)but-3-en-2-yl acetate (**2d**): Yield 98% (2.6 g), spectroscopic data consistent with those reported in the literature [68].

(*E*)-4-phenylbut-3-en-2-yl propionate (**3a**): Yield 96% (2.9 g), spectroscopic data consistent with those reported in the literature [69].

(*E*)-4-(4′-methylphenyl)but-3-en-2-yl propionate (**3b**): Yield 97% (2.9 g), spectroscopic data consistent with those reported in the literature [69].

(*E*)-4-(2’,5’-dimethylphenyl)but-3-en-2-yl propionate (**3c**): Yield 98% (3.0 g), spectroscopic data consistent with those reported in the literature [63].

(*E*)-4-(4’-methoxyphenyl)but-3-en-2-yl propionate (**3d**): Yield 97% (2.8 g), spectroscopic data consistent with those reported in the literature [70].

(*E*)-4-phenylbut-3-en-2-yl butyrate (**4a**): Yield 96% (2.7 g), spectroscopic data consistent with those reported in the literature [71].

(*E*)-4-(4′-methylphenyl)but-3-en-2-yl butyrate (**4b**): Yield 98% (2.9 g), spectroscopic data consistent with those reported in the literature [71].

(*E*)-4-(2,5-dimethylphenyl)but-3-en-2-yl butyrate (**4c**): Yield 97% (3.1 g), colorless liquid, $n_{D}^{20}$ = 1.5175; ^1^H NMR, δ: 0.96 (t, *J* = 7.8 Hz, 3H, CH_3_-8), 1.42 (d, *J* = 6.6 Hz, 3H, CH_3_-1), 1.68 (sextet, *J* = 7.8 Hz, 2H, CH_2_-7), 2.30, 2.31 (two s, 6H, CH_3_-9 and CH_3_-10), 2.29–2.32 (m, 2H, CH_2_-6), 5.55 (m, 1H, H-2), 6.06 (dd, *J* = 15.6 and 6.6 Hz, 1H, H-3), 6.79 (d, *J* = 15.6 Hz, 1H, H-4), 6.98 (m, 1H, H-4′), 7.03 (d, *J* = 7.8 Hz, 1H, H-3′), 7.25 (s, 1H, H-6′); ^13^C NMR, δ: 13.6 (C-8), 18.5 (C-7), 19.2 (C-9), 20.9 (C-10), 20.5 (C-1), 36.6 (C-6), 70.9 (C-2), 126.2 (C-6′), 128.5 (C-4′), 129.4 (C-4), 130.0 (C-3), 130.2 (C-3′), 132.6 (C-2′), 135.2, (C-1′), 135.4 (C-5′), 172.9 (C-5); IR (cm^−1^): 1735 (s), 1457 (m), 1182 (s), 1040 (s), 966 (s), 808 (m); HRMS: calcd for C_16_H_22_O_2_ [M + Na]^+^: 269.1517, found 269.1518.

(*E*)-4-(4′-methoxyphenyl)but-3-en-2-yl butyrate (**4d**): Yield 98% (3.0 g), spectroscopic data consistent with those reported in the literature [71].

### 3.4. Determination of Enzyme Activity

The enzymatic activity of the free and immobilized preparations of Lecitase™ Ultra was determined according to the procedure described by Pencreac’h and Baratti [72] with some modifications, based on the amount of p-nitrophenol released during the hydrolysis of p-nitrophenyl palmitate. The reaction mixture consisted of 75 µL of 1 mM p-NPP, 5.75 mL of 10 mM Tris/HCl buffer (pH 8.0), and 50 µL of free lipase solution or 30 mg of immobilized lipase. The mixture was kept at 37 °C for 130 min and the reaction was terminated by the addition of cold ethanol. The absorbance of released p-nitrophenol was measured on a Cintra 101 spectrophotometer, GBC Scientific Equipment at 410 nm. One international unit of *p*-NPP activity (1 U) was defined as the amount of enzyme necessary to produce 1 µmol of *p*-NP per minute under the conditions described above.

### 3.5. Immobilization of Lecitase™ Ultra by Entrapment in Calcium Alginate

Procedure of immobilization described by Blandino et al. was used with some modifications [73]. Lecitase™ Ultra (2.2 mL) was added to 10 mL of 6% of sodium alginate solution and stirred for 1 h on a magnetic stirrer. The mixture was dropped using a syringe into 500 mL of 0.3 M CaCl_2_. Formed beads of immobilized preparation were left for 1 h in the solution. Finally, the preparation was filtered, washed with distilled water to remove the residual CaCl_2_, and freeze-dried for 24 h using an LYO GT2-Basic lyophilizer at 10 mbar. As a result, 0.55 g of Lecitase™ Ultra immobilized in calcium alginate (LU-CA) was obtained and stored at 4 °C until use.

### 3.6. Immobilization of Lecitase™ Ultra by Adsorption on Polyacrylic Resin (Supelite™ DAX-8)

Immobilization of the enzyme was carried out using the method described by Egger et al. [74] with some modifications. Polyacrylic resin Supelite™ DAX-8 (20 g) was washed with Tris-HCl buffer and distilled water (4 × 20 mL). The carrier was left at room temperature for 24 h, and 1 g of pretreated support was placed in a twisted 20 mL vial containing 10 mL of Tris-HCl buffer and 4 mL of Lecitase™ Ultra. The mixture was shaken on a laboratory end-over-end shaker for 4 days. Immobilized enzyme was filtered, washed with Tris-HCl buffer (2 × 10 mL) and freeze-dried as described in paragraph 3.5 (24 h). Finally, 0.8 g of Lecitase™ Ultra immobilized on Supelite ™ DAX-8 (LU-DAX) was obtained and stored at 4 °C until use.

### 3.7. Immobilization of Lecitase™ Ultra by Covalent Bonds on Cyanogen Bromide-Activated Agarose (CNBr)

Immobilization of the enzyme was carried out according to the modified protocol reported by Dos Santos [36]. Cyanogen bromide-activated agarose (3 g) was successively washed on a Schott G3 funnel with 600 mL of cold 1 mM HCl in several portions for 30 min and then with 50 mL of coupling buffer (0.1M NaHCO_3_, 0.5 M NaCl, pH 8.3). Commercial Lecitase™ Ultra (2 mL) in a 250 mL Erlenmayer flask was diluted in 12 mL of coupling buffer and 3 g of prepared wet carrier was added. The mixture was shaken on a rotary shaker for 1 h at room temperature, filtered on a Schott G3 funnel, and rinsed with 50 mL of coupling buffer. The immobilized enzyme was transferred back to the Erlenmayer flask and incubated with 50 mL of 1 M ethanolamine (pH 8.0) at room temperature for 2 h, filtered on a Schott G3 funnel, then washed with acetate buffer (pH 4.0, 4 × 10 mL) and finally with 50 mL of coupling buffer. After freeze-drying (24 h), 2.8 g of Lecitase™ Ultra immobilized on cyanogen bromide-activated agarose (LU-CNBr) was obtained and stored at 4 °C until use.

### 3.8. Immobilization of Lecitase™ Ultra by Covalent Bonds on Modified Bacterial Cellulose (MBC)

Bacterial cellulose spheres produced by shaking cultures of *Komagataeibacter xylinus* and modified with polyethyleneimine and ferromagnetic particles were used as a carrier for enzyme immobilization. The detailed procedure of the carrier preparation as well as Lecitase™ Ultra immobilization has been described earlier [50]. Briefly, prior to immobilization, the carrier was activated with 1% glutaraldehyde solution in 100 mM phosphate buffer (pH 7.0) by shaking at a roller shaker. After activation, the modified bacterial cellulose (MBC) was collected using a magnetic separator and rinsed with phosphate buffer to remove an excess of glutaraldehyde. Next, 10 mL of enzyme solution was added to 5 mL of activated carrier and the mixture was incubated at 4 °C at a roller shaker. After 24 h, the immobilized enzyme was collected using a magnetic separator and the supernatant was removed. The preparation was washed with phosphate buffer and incubated at 4 °C for 1 h with NaBH_4_ solution in the phosphate buffer. Finally, 4 g of Lecitase™ Ultra immobilized on modified bacterial cellulose (LU-MBC) was collected using a magnetic separator and flushed successively with phosphate buffer containing 100 mM NaCl, 0.25% Triton-X100, and phosphate buffer. The preparation was freeze-dried for 24 h and stored at 4 °C until use.

### 3.9. Hydrolysis of Esters **2a–d**, **3a–d** and **4a–d** Catalyzed by free or Immobilized Lecitase™ Ultra

#### 3.9.1. General Procedure

A total of 0.06 U of free enzyme Lecitase™ Ultra (2 mL) or 0.06 U of immobilized preparation of this enzyme (1.33 g LU-MBC or 1.15 g LU-DAX-8, or 1.56 g LU-CA or 1.05 g LU-CNBr) and 0.2 g of ester dissolved in 0.5 mL of organic solvent (acetone, DMF or DMSO) were placed in 10 mL screw cap glass vials containing 3.5 mL of the phosphate buffer (pH 7.2) or Tris/HCl buffer (pH 8.2). Vials were shaken at 750 rpm on a magnetic stirrer at 20 °C. Samples (0.6 mL) of the reaction mixture were taken at different time intervals and extracted with diethyl ether (2 × 2 mL). In the case of the reaction with the immobilized enzyme, the extracts were filtrated through diatomaceous earth (Celite 560) to completely remove an immobilized biocatalyst. The extracts were dried and solvent was removed by evaporation in vacuo. Before chiral gas chromatography (CGC) samples taken from the reaction mixture were treated with propionyl or acetyl chloride to derivatize inseparable enantiomers of alcohols **1a–d** into the corresponding esters as described earlier [60].

#### 3.9.2. The Effect of Dosage of Lecitase Ultra immobilized Cyanogen Bromide-Activated Agarose (LU-CNBr)

Different dosages of LU-CNBr and (*E*)-4-phenylbut-3-en-2-yl acetate **2a** (0.2g) in 0.5 mL of acetone were placed in 10 mL screw cap glass vials containing 3.5 mL of the phosphate buffer (pH 7.2). Vials were shaken at 750 rpm on magnetic stirrer for 24 h. Samples (0.6 mL) from the reaction mixture were taken after 24 h, filtrated through Celite 560, and prepared for chiral GC as described in paragraph 3.9.1.

#### 3.9.3. Isolation of Products Obtained by Hydrolysis of Esters **4a**–**4c**—General Procedure

Hydrolysis of butyrate **4a**–**4c** (0.2 g) in the phosphate buffer was carried out as described in paragraph 3.9.2 using LU-CNBr (0.06 U). After 24 h of reaction, the products were extracted with ether diethyl (2 × 5 mL) and the combined organic layers were filtered through Celite 560, washing the adsorbent with 8 mL of diethyl ether. The extract was dried over anhydrous magnesium sulfate, the solvent was evaporated in vacuo, and the mixture of products was separated by preparative TLC (hexane: acetone, 10:1).

After hydrolysis of ester **4a** (0.2 g, 0.9 mmol) the following products were obtained:

(*S,E*)-4-phenylbut-3-en-2-yl butyrate ((*S*)-**4a**): yield 45% (0.09 g), *ee* 96%, ${\left[\alpha \right]}_{D}^{20}$ = −51.3 (c 0.36; CHCl_3_); lit. [71]: ${\left[\alpha \right]}_{D}^{20}$ = −111.3 (c 1.0; CHCl_3_, *ee* 97%).

(*R,E*)-4-phenylbut-3-en-2-ol ((*R*)-**1a**): yield 47% (0.063 g), *ee* 97%, ${\left[\alpha \right]}_{D}^{20}$ = +19.6 (c 1.35; CH_2_Cl_2_); lit. [65]: ${\left[\alpha \right]}_{D}^{20}$ = +19.9 (c 1.6; CH_2_Cl_2_, *ee* 96%)

After hydrolysis of ester **4b** (0.2 g, 0.8 mmol) the following products were isolated: (*S,E*)-4-(4′-methylphenyl)but-3-en-2-yl butyrate ((*S*)-**4b**): yield 45% (0.089 g), *ee* 89%, ${\left[\alpha \right]}_{D}^{20}$ = −108.1 (c 0.39; CHCl_3_), lit. [71]: ${\left[\alpha \right]}_{D}^{20}$ = −115.9 (c 1.0; CHCl_3_, *ee* 98%)

(*R,E*)- 4-(4′-methylphenyl)but-3-en-2-ol ((*R*)-**1b**): yield 47% (0.066 g), *ee* 98%, ${\left[\alpha \right]}_{D}^{20}$ = +22.7 (c 1.22; CH_2_Cl_2_), lit [69]: ${\left[\alpha \right]}_{D}^{20}$ = +22.5 (c 1.5; CH_2_Cl_2_, ee 96%)

Hydrolysis of ester **4c** (0.2 g, 0.8 mmol) afforded the following products: (*S,E*)-4-(2′,5′-dimethylphenyl)but-3-en-2-yl butyrate ((*S*)-**4c**): yield 48% (0.096 g), *ee* 99%, ${\left[\alpha \right]}_{D}^{20}$ = −46.2 (c 0.38; CH_2_Cl_2_).

(*R,E*)-4-(2′,5′-dimethylphenyl)but-3-en-2-ol ((*R*)-**1c**): yield 43% (0.062 g), *ee* 99%, ${\left[\alpha \right]}_{D}^{20}$ = +16.5 (c 0.37; CH_2_Cl_2_); lit. [63]: ${\left[\alpha \right]}_{D}^{20}$ = +16.2 (c 1.7; CH_2_Cl_2_, *ee* 98%).

## 4. Conclusions

Results of this survey allowed for the extension of the scope of application for enzyme Lecitase™ Ultra in the asymmetric synthesis. This biocatalyst proved to be an attractive alternative to the commonly used commercial lipase preparations in the kinetic resolution of 4-arylbut-3-en-2-yl esters through their enantioselective hydrolysis. In the reactions catalyzed by the free enzyme, very good kinetic resolution was obtained in reactions carried out in the phosphate buffer, with the addition of acetone as a co-solvent. The enantioselectivity of hydrolysis increased along with the acyl chain length. The significant increase in the optical purity of unreacted substrates and products (up to 90–99%) as well as the reduction of a reaction time necessary for optimal kinetic resolution was achieved using immobilized preparations (up to 24–48 h). Particularly effective was the enzyme immobilized on cyanogen bromide-activated agarose (LU-CNBr). This preparation also catalyzed effective kinetic resolution of substrates containing methyl substituents on phenyl ring (**4b,4c**), however, it did not show any enantioselectivity toward 4-(4’-methoxyphenyl)but-3-en-2-yl butyrate (**4d**).

Further research will be aimed at the extension of the scope of the applicability of Lecitase™ Ultra not only in the hydrolysis of structurally diverse esters, but also in the kinetic resolution of alcohols through their transesterification.

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
