# Peer review of "Free and Immobilized Lecitase™ Ultra as the Biocatalyst in the Kinetic Resolution of (E)-4-Arylbut-3-en-2-yl Esters"

_molecules, 2020, doi:10.3390/molecules25051067_

Round 1

Reviewer 1 Report

Leśniarek and colleagues have investigated the use of Lecitase Ultra for the kinetic resolution of racemic 4-phenylbut-3-en-2-yl esters. The work has been carried out adequately, and the conclusions will be of interest to others working in the field. Specifically, the type of buffer, type of co-solvent, free enzyme in solution versus immobilized enzyme, as well as the influence of the acyl chain length were studied. The experiments showed that phosphate buffer, with acetone as co-solvent, and Lecitase Ultra immobilized on cyanogen-bromide activated agarose gave the best results.

Unfortunately, the writing style of the manuscript is somewhat verbose, which obscures the clarity. Below I give a suggestion to improve the abstract. I leave it to the authors to improve/shorten the remainder of the manuscript.

The influence of buffer type, co-solvent type, and acyl chain length was investigated for the enantioselective hydrolysis of racemic 4-arylbut-3-en-2-yl esters using Lecitase Ultra. Immobilized preparations of the Lecitase Ultra enzyme had significantly higher activity and enantioselectivity than free enzyme, particularly for 4-phenylbut-3-en-2-yl butyrate as substrate. Moreover, the kinetic resolution with immobilized enzyme was achieved in a much shorter time (24-48 h). Lecitase Ultra, immobilized on cyanogen bromide-activated agarose, was particularly effective, producing, after 24 h of reaction time in phosphate buffer (pH 7.2) with acetone as co-solvent, both (R)-alcohols and unreacted (S)-esters with excellent enantiomeric excesses (ee 90-99%). These conditions and enzyme were also suitable for the kinetic separation of racemic (E)-4-phenylbut-3-en-2-yl butyrate analogs containing methyl substituents on the benzene ring, but they did not show any enantioselectivity towards (E)-4-(4'-methoxyphenyl)but-3-en-2-yl butyrate.

(131 words instead of 209; 1034 characters instead of 1552)

Author Response

We would like to thank the Reviewer for a great appreciation of our work and for  his valuable critical comments on the style of the manuscript which helped us to improve the quality of our paper. According to the Reviewer’s suggestions we corrected the manuscript and made some changes in the text which are highlighted in yellow.

Specific remarks:

Reviewer: "Unfortunately, the writing style of the manuscript is somewhat verbose, which obscures the clarity. Below I give a suggestion to improve the abstract. I leave it to the authors to improve/shorten the remainder of the manuscript"

Response: We improved the Abstract according to the suggestion of Reviewer 1 and 2 (lines 13-24). Indeed, it is now more concise and shortened. Likewise, we also reformulated some fragments of manuscript (lines 105-111, 138-147, 441-455) to make the discussion and conclusions more clear and relevant.

I hope that our corrections and explanations will be sufficient for acceptation of the paper.

With kind regards

W.Gładkowski

Reviewer 2 Report

Some corrections were suggested to the authors. In some places, the text should be rewritten for a clearer and more relevant discussion. 

Only one important demand was adresed to the authors, regarding the results presented in Figure 1 - Figure 3.

I recommend the acceptance of this article after some minor revision (see the corrections and comments on the attached file).

16-14, 24, 26-28, 30, 54, 56, 58, 73-74, 84, 86, 100-102, 104, 111-112, 114-125, 146, 152-170, 183-185, 186-187, 197, 199, 202, 216, 233, 277, 318, 377, 385, 394, 406, 427, 430, 462.

Author Response

We would like to express our thanks to Reviewer for the valuable comments. According to the Reviewer’s suggestions we corrected the manuscript and made some changes in the text which are highlighted in yellow. Here we would like to comment on the changes we have made in the manuscript and give explanations of some issues that had been raised by Reviewer.

I hope that our corrections and explanations will be sufficient for acceptation of the paper.

Reviewer: General remarks:

"Some corrections were suggested to the authors. In some places, the text should be rewritten for a clearer and more relevant discussion" I recommend the acceptance of this article after some minor revision (see the corrections and comments on the attached file". Only one important demand was addressed to the authors, regarding the results presented in Figure 1 - Figure 3".

Figure 1-Figure 3 Reviewer: "here is a problem with the results after 24 h when LU-CNBr was used: the ester ee is high, while the alcohol ee decreased, since in 24 h the conversion exceed 50%. For this experiments, the process must be studied also at shorter reaction time, trying to stop it at approx. 50% conversion, when both substrate (ester) and product (alcohol) are probably almost pure (ee high). A good example is represented by LU-MBC, when after 48 h the conversion and ee are all near the maximal values"

Response: We measured  ee of substrate and product also in shorter time; for example for LU-CNBr-catalyzed hydrolysis of acetate 2a when the conversion was approaching 50% the situation was opposite: ee of alcohol was admittedly higher (96-97%) but the ee of ester was lower (90%) - that's why we had to find the compromise between these values and decided to give only one result, after 24 h, when ee of substrate was very high (99%). In each case we would like to emphasize the results in which very high ee of substrate or product  or two of them are obtained. Similar situation was for LU-MCB in the hydrolysis of butyrate 4a when optically pure ester was obtained (Figure 3). In the hydrolysis of propionate 3a even when the reaction exceeded 50% conversion, the highest ee of unreacted ester was 90%, together with lower ee of alcohol. (Figure 2).

In order to present clear course of the reaction, in the revised version of manuscript, we completed the discussion of results contained in Figures 1-3, taking into account the conversion values as indicated by Reviewer. It is given below:

lines 178-182: "A highly enantioselective (E=99) was also the reaction catalyzed by the LU-CNBr, in this case after 24 h the conversion exceeded 50% which resulted in lower ee of alcohol 1a (90%) but excellent ee of acetate 2a (99%). At shorter reaction time (20h, data not shown), when the conversion was close to 50%, ee of alcohol was higher, but lower ee of acetate was observed."

lines 200-202: "Accordingly, the enantiomers of substrate were resolved in a significantly shorter time (24 h) and at 53% conversion alcohol 1a and unreacted ester 3a were obtained with enantiomeric excesses of 79% and 90% respectively"

lines 217-218:".....in this case after 24 h the conversion exceeded 50% and it was possible to obtain optically pure unreacted butyrate 4a."

Other corrections:

Lines 13-24. We corrected the Abstract according to suggestion of Reviewer 1 and 2

Line 25: according to suggestion, "chiral alcohols" has been specified to " (E)-4-arylbut-3-en-2-ols"

Line 49: The fragment "esters of mandelic acid and esters of other structurally related 2-hydroxy carboxylic acids containing benzene ring" has been reformulated to "esters of 2-hydroxyacids with phenyl ring, including mandelates" according to suggestion of Reviewer

Line 50: Reviewer: "you should explain/justify why it is interesting! otherwise is only your opinion"

Response: In this sentence we just wanted to differentiate between examples of kinetic resolution given in lines 49-50 and regioselective hydrolysis of prochiral compound. In order to avoid misunderstanding, we changed "interesting example" to "another example" in the revised version.

Lines 53-57: Reviewer: "reformulate the sentence: Immobilization of enzymes on different supports is one of the most common techniques to save time at the stage of a separation the biocatalyst from the reaction environment"

Response: We agree that this sentence could not be clear - we would like to stress the benefits of immobilization in context of the separation of biocatalyst after reaction and its reuse. To make this more clear, we changed this fragment as follows:

"Immobilization of enzymes on different supports is a useful tool to increase their stability, activity and selectivity under operational conditions [42–44]. Because of a simpler separation of the biocatalyst from the reaction environment, such preparations may be reused in many catalytic cycles without loosing their properties which increases the economic efficiency of the industrial process."

Lines 60-61 Reviewer: "entrapment probably in both cases, gelatin and alginate"

Response: Indeed, "entrapment" is more precise in this context. According to suggestion we changed this fragment as follows:

 ".....by entrapment in gelatine hydrogel [47] or in different calcium alginates......"

Lines 68-70 Reviewer: Reformulate the entire sentence: "In the Lecitase Ultra-catalyzed kinetic resolutions of racemates reported so far its ability to enantioselective hydrolysis of ester bond has been limited mainly to the substrates presenting stereogenic center in the acyl donors"

Response: The sentence was reformulated as follows:

"In the previous investigations, kinetic resolutions of racemic esters through their Lecitase Ultra-catalyzed enantioselective hydrolysis has been limited mainly to the esters of chiral carboxylic acids"

Lines 79-81 The sentence has been corrected according to Reviewer's suggestion as follows:

"Here we present the results of our further investigations in this field, aiming to increase the process efficacy through optimization of some reaction conditions (pH of the reaction medium, addition of organic co-solvent) as well as by enzyme immobilization."

Lines 91-93 The sentence has been corrected according to Rewiewer's suggestion as follows:

"Studies towards kinetic resolution of prepared compounds were started for the free Lecitase Ultra (LU)-catalyzed hydrolysis of model 4-phenylbut-3-en-2-ol esters (2a-4a) to investigate the influence of the length of an acyl chain, type of a co-solvent, and type of a buffer on the process stereoselectivity (Table 1)."

Lines 94 The headline of Table 1 was shortened according to Reviewer's suggestion

Table 1 Reviewer: "pH of all buffers should be mentioned in this column"

Response: According to the reviewer's suggestion, pH values of Tris-HCl and phosphate buffer have been added as footnotes 3 and 4 under the Table.

Lines 103-104 We made the corrections suggested  by Reviewer

Lines 105-111 Reviewer: "Please try to present the results more clear and condensed, based on the Table 1 results"

Response: Following Reviewer's advice, we reformulated and shortened this fragment of discussion, to present it more clearly. Now it is as follows:

"Higher enantiomeric excess of alcohol 1a and shorter reaction time necessary to achieve the best resolution was observed with the elongation of acyl chain. The best results were obtained for butyrate 4a; in this case the highest enantioselectivity (E>200) was observed after 96 h of reaction, resulting in the optically pure product (Entry 6). The positive effect of acyl chain lenght of hydrolyzed esters 2a-4a on their kinetic resolution was a constant tendency, observed also in experiments carried out in Tris-HCl (Entries 1-3) as well as using other co-solvents (Entries 7-9 and 10-12)."

Headline of Table 2: Reviewer: "phenyl or aryl is more correct"

Response: We changed the headline according to suggestions.

Lines 138-147 Reviewer: "please reformulate the entire paragraph as earlier described in line 113"

Response: The paragraph was shortened to contain only the most important comments on the results from Table 2. Now it sounds as given below:

“Based on the previous results of kinetic resolution of esters 2a-4a, the procedure using phosphate buffer as reaction medium and the addition of acetone as a co-solvent was applied Regardless of the acyl chain length, esters of (E)-4-(4’-methoxyphenyl)but-3-en-2-ol (2d-4d) were hydrolyzed with no enantioselectivity (Entries 7-9), and for esters containing methyl groups (2b-4b, 2c-4c) lower enantioselectivities of the reactions were observed (E≤18) compared to (E)-4-phenylbut-3-en-2-yl esters 2a-4a (Table 1, entries 4-6). The beneficial effect of elongation of acyl chain on the results of kinetic resolution was confirmed only in the group of esters with the p-methyl substituted phenyl ring (2b-4b). Better resolution was observed for compounds with 2,5-dimethylsubstituted phenyl ring than for those with the p-methylphenyl substituent, which was particularly noticeable in the case of acetates and propionates.

Lines 160-162 Reviewer: "Reformulate the sentence"

Response: It was reformulated as follows:

"Preparation LU-CNBr was previously used to the production of optically pure (S)-3-methyl phenylglutarate through asymmetric hydrolysis of dimethyl 3-phenylglutarate (48 h, 80% yield)."

Headline of Table 3: We changed the "Characteristic" to "Activity" according to Reviewer's suggestion

Lines 175-176 Reviewer: "Redundant again: process enantioselectivity depends on the ee of substrate and products, which are depending on the reaction  conversion."

Response: We replaced fragment: ".....the enantioselectivity of the reaction and the increase of enantiomeric excesses of unreacted ester and produced alcohol" to "parameters of kinetic resolution".

Line 177: we inserted reference to  Figure 1 in the text as requested by Reviewer

Line 260-267: Reviewer: "it is normal, the reaction rate decreased with the enzyme quantity, without to affect the stereoselectivity" Have you tried a higher dosage, in order to reduce the reaction time? When an immobilized and recyclable enzyme is used, this fact can be used in order to reduce the reaction time!

Response: Thank you very much for this valuable comment. We did not try the higher dosage of enzyme to reduce reaction time, we just wanted to check what amount of enzyme is necessary for effective resolution after 24 h of reaction. In our future investigations we plan to make experiments to clarify how many cycles we can carry out without significant decrease of the activity of enzyme

Line 355 The title of paragraph 3.5 was completed according to suggestion and now it sounds: "Immobilization of Lecitase Ultra by entrapment in calcium alginate"

Line 363 The title of paragraph 3.6 was completed according to suggestion and now it sounds: "Immobilization of Lecitase Ultra by adsorption on polyacrylic resin (SupeliteTM DAX-8)"

Line 372 The title of paragraph 3.7 was completed according to suggestion and now it sounds: "Immobilization of Lecitase Ultra by covalent bonds on cyanogen bromide-activated agarose (CNBr)"

Line 384 The title of paragraph 3.8 was completed according to suggestion and now it sounds: "Immobilization of Lecitase Ultra by covalent bonds on modified bacterial cellulose (MBC)"

Line 368 Reviewer: Have you tried to reduce this time? Are you sure that the immobilization yield increase in time, taking into account that the enzyme adsorption on this hydrophobic support is a reversible one and the equilibrium is probably established in shorter time.

Response: During the immobilization process, the samples were taken and the activity of enzyme was measured. The maximum of activity of enzyme immobilized on resin we have at our diposal was determined after 4 day of immobilization  and therefore we chose this time as optimal for immobilization

Line 405 Reviewer: "please add the approx. temperature, since it is a very important parameter of EKR processes"

Response: "The temperature 20 °C has been added"

Line 408 The term "immobilized preparation" was changed to "immobilized biocatalyst" according to Reviewer's suggestion

Lines 441-455, Conclusions

 Reviewer: "It could be rewritten and shortened, pointing the most relevant results"

Response: The conclusion section was shortened according to the Reviewer's suggestion. Now it sounds as follows:

" Results of this survey allowed for the extension of the scope of application for enzyme Lecitase Ultra in the asymmetric synthesis. This biocatalyst proved to be attractive alternative to the commonly used commercial lipase preparations in the kinetic resolution of 4-arylbut-3-en-2-yl esters through their enantioselective hydrolysis. In the reactions catalyzed by free enzyme very good kinetic resolution was obtained in reactions carried out in the phosphate buffer, with the addition of acetone as a co-solvent. The enantioselectivity of hydrolysis increased along with the acyl chain lenght. The significant increase of optical purity of unreacted substrates and products (up to 90-99%) as well as reduction of a reaction time necessary for optimal kinetic resolution was achieved using immobilized preparations (up to 24-48 h). Particularly effective was the enzyme immobilized on cyanogen bromide-activated agarose (LU-CNBr). This preparation catalyzed also effective kinetic resolution of substrates containing methyl substituents on phenyl ring (4b,4c), however it did not show any enantioselectivity towards 4-(4'-methoxyphenyl)but-3-en-2-yl butyrate (4d).

                Further research will be aimed at the extension of the scope of applicability of Lecitase Ultra not only to the hydrolysis of structurally diversed esters but also to the kinetic resolution of alcohols through their transesterification."

With kind regards

W.Gładkowski